

# Peptide-based NTA(Ni)-nanodiscs for studying membrane enhanced FGFR1 kinase activities

Juanjuan Liu[1,2], Lei Zhu[1], Xueli Zhang[3,4], Bo Wu[1], Ping Zhu[3,4], Hongxin Zhao[1] and Junfeng Wang[1,2,5]

[1] High Magnetic Field Laboratory, Key Laboratory of High Magnetic Field and Ion Beam Physical Biology, Hefei Institutes of Physical Science, Chinese Academy of Sciences, Hefei, Anhui, China
[2] University of Science and Technology of China, Hefei, Anhui, China
[3] National Laboratory of Biomacromolecules, CAS Center for Excellence in Biomacromolecules, Institute of Biophysics, Chinese Academy of Sciences, Beijing, China
[4] University of Chinese Academy of Sciences, Beijing, China
[5] Institute of Physical Science and Information Technology, Anhui University, Hefei, Anhui, China

## ABSTRACT

Tyrosine autophosphorylation plays a crucial regulatory role in the kinase activities of fibroblast growth factor receptors (FGFRs), and in the recruitment and activation of downstream intracellular signaling pathways. Biophysical and biochemical investigations of FGFR kinase domains in membrane environments offer key insights into phosphorylation mechanisms. Hence, we constructed nickel chelating nanodiscs based on a 22-residue peptide. The spontaneous anchoring of N-terminal His$_6$-tagged FGFR1c kinase domain (FGFR1K) onto peptide nanodiscs grants FGFR1K orientations occurring on native plasma membranes. Following membrane incorporation, the autophosphorylation of FGFR1K, as exemplified by Y653 and Y654 in the A-loop and the total tyrosine phosphorylation, increase significantly. This in vitro reconstitution system may be applicable to studies of other membrane associated phenomena.

## INTRODUCTION

As members of the receptor tyrosine kinases (RTKs) family, fibroblast growth factor receptors (FGFRs) play important roles as regulators of proliferation, differentiation, migration, and survival in various cell types (*Eswarakumar, Lax & Schlessinger, 2005*; *Mikhaylenko et al., 2018*). Accordingly, deregulation of FGFR signaling contributes to various pathological conditions and developmental syndromes (*Tiong, Mah & Leong, 2013*; *Turner & Grose, 2010*). FGF ligands bind the extracellular domain of FGFR with facilitating cofactors, such as heparin sulfate and klothos (*Goetz et al., 2012*; *Schlessinger et al., 2000*), and induce receptor homo-dimerization and subsequent activation of cytoplasmic tyrosine kinase domains by tyrosine autophosphorylation at multiple sites. At least seven tyrosine autophosphorylation sites have been described in FGFR1 (Y463, Y583, Y585, Y653, Y654, Y730, and Y766; Fig. 1A) (*Mohammadi et al., 1996*). Phosphorylation

Corresponding authors
Hongxin Zhao, zhx@hmfl.ac.cn
Junfeng Wang, junfeng@hmfl.ac.cn

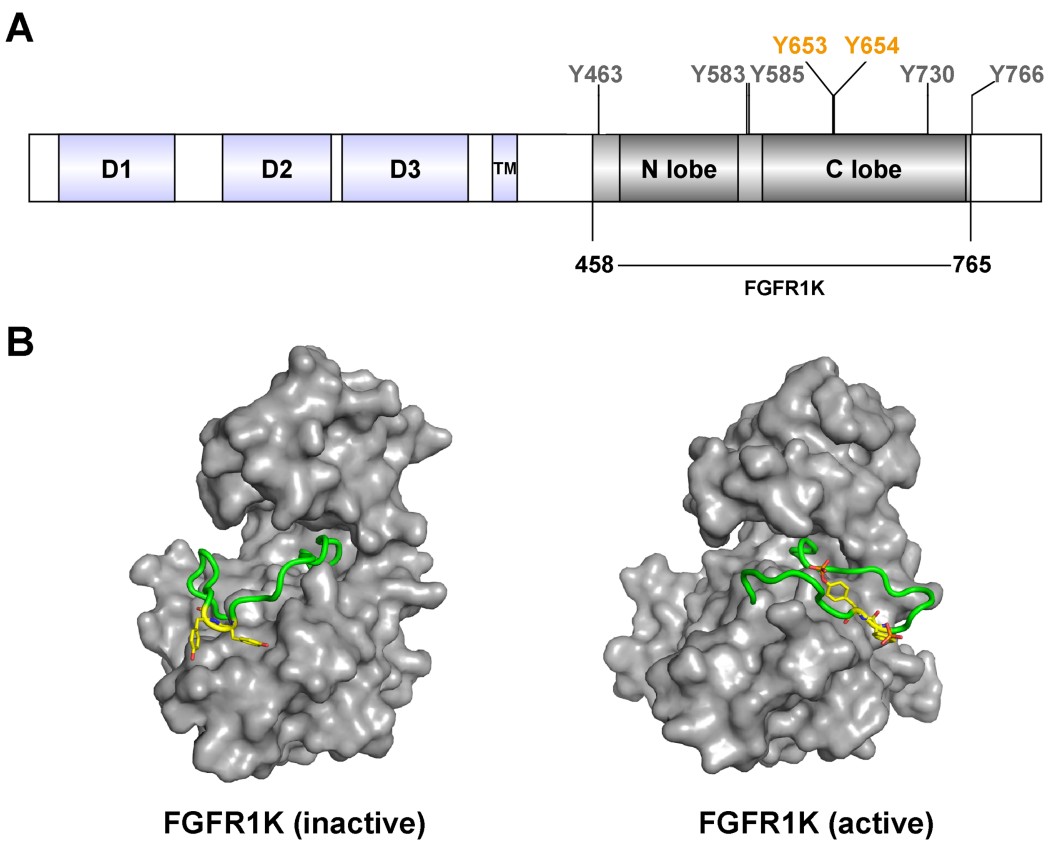

**Figure 1  FGFR1c kinase (FGFR1K) and phosphorylation sites.** (A) FGFR1c consists of three domains: extracellular receptor domain (D1, D2, D3), transmembrane domain (TM) and kinase domain (N lobe and C lobe). The kinase domain posesses seven tyrosine autophosphorylation sites, including Y653 and Y654 in the active loop (A-loop). The kinase domain (residues 458-765; FGFR1K) were constructed and recombined expressed for functional studies. (B) Surface representation of inactive (left, PDB ID: 3KY2) and active (right, PDB ID: 3GQI) conformations of the FGFR1 kinase domain. The A-loop is shown as a cartoon representation in green, and Y653 and Y654, which are unphosphorylated and phosphorylated in the respective inactive and active forms, are highlighted as yellow sticks.

of the A-loop residues Y653 and Y654 (highlighted in Fig. 1B) (*Bae et al., 2010*, *2009*) is critical for upregulation of kinase activity (*Furdui et al., 2006*). Moreover, phosphorylation sites of tyrosine residues in other parts of the cytoplasmic region serve as specific binding sites for downstream signaling molecules containing Src Homology 2 (SH2) or phospho-tyrosine binding domains, resulting in the activation of specific signaling pathways (*Babina & Turner, 2017*; *Bae et al., 2009*).

To investigate the mechanisms of RTK autophosphorylation, many biological and biophysical studies have been performed using analyses of isolated kinase domains in aqueous solution (*Klein et al., 2015*; *Kobashigawa et al., 2015*). Yet other studies of RTKs show that isolated kinase domains fail to reproduce in vivo observations (*Bae et al., 2010*). This largely reflects the importance of cell membrane localization of kinase domains, which is essential for promoting receptor dimerization and cooperative cross-receptor autophosphorylation.

Given the importance of protein-membrane interactions, accurate experimental mimicry of membrane anchors is necessary for studies of RTK kinase domains. A strategy for binding poly-histidine-tagged proteins to small unilamellar vesicles or liposomes was previously developed using nickel chelating lipids (*Esposito, Shrout & Weis, 2008*; *Zhang et al., 2006*). Weis and coworkers pioneered the use of this strategy in their studies of a prokaryotic signal transduction system that mediates bacterial chemotaxis (*Montefusco et al., 2007*; *Shrout, Montefusco & Weis, 2003*). In addition, Monsey and coworkers used nickel chelating liposomes to form heterodimers of epidermal growth factor receptors between Her4 and Her2/neu tyrosine kinase domains mimicking the in vivo behaviors of full-length receptors following ligand binding. Their study showed that addition of nickel chelating liposomes to the Her4 kinase domain resulted in a 40-fold increase in kinase activity (*Monsey et al., 2010*).

Nanodiscs are discoidal nanomembrane particles with a planar phospholipid bilayer enwrapped by proteins such as apolipoprotein A-I or membrane scaffolding proteins (MSP) (*Parmar et al., 2016*). Nanodiscs have been widely used for analyzing structures and functions of membrane proteins by dispersing them in solution (*Hagn, Nasr & Wagner, 2018*; *Nasr et al., 2017*; *Rouck et al., 2017*). In addition, the lipid compositions of nanodiscs can be controlled precisely providing a nanoscale membrane surface for investigating various membrane recognition events (*Denisov & Sligar, 2017*). To simplify the preparation procedure at the same time maintaining the stabilization provided by nanodiscs, a new strategy, named peptide nanodisc, has been developed. This is based on the use of synthetic peptides mimicking the amphipathic helices of Apolipoprotein I, whereas the nanodisc sizes can be conveniently controlled by varying the lipid/peptide ratio (*Imura et al., 2014*; *Miyazaki et al., 2010*).

Using a 22 amino-acid peptide derived from Apolipoprotein I protein (*Zhang et al., 2016*), we herein introduced nickel chelating lipids into peptide nanodisc to generate a nanoscale 2D-membrane mimics for studying FGFR1 kinase domain (FGFR1K) activity. Our data show that histidine-tagged FGFR1K can be conveniently assembled onto this peptide-based nickel chelating nanodiscs. The autophosphorylation of tyrosines, especially Y653 and Y654 on the activated loop of kinase region, will be studied using nanoscale membrane templates.

## MATERIALS AND METHODS

### Chemicals and reagents

1, 2-dimyristoyl-sn-glycero-3-phosphocholine (DMPC) and 1, 2-dioleoyl-sn-glycero-3-[(N-(5-amino-1-carboxypentyl) iminodiacetic acid) succinyl] (nickel salt) (DGS-NTA (Ni)) were purchased from Avanti Polar Lipids. Antibodies against FGF receptor 1 (D8E4; cat. 9740), FGFR phospho-Tyr-653/4 (cat. 3476), and phospho-tyrosine (P-Tyr-1000; cat. 8954) were purchased from Cell Signaling Technology, Inc. Antibodies against FGFR1 phospho-Tyr-653 (cat. ab173305) and phospho-Tyr-654 (cat. ab59194) were purchased from Abcam. Dithiobis (succinimidyl propionate) (DSP; cat. C110213) was purchased from Sangon Biotech. The 22-residue peptide PVLDLFRELLNELLEALKQKLK was synthesized by GL Biochem (Shanghai) Ltd., Shanghai, China.

## FGFR1K expression and purification

DNA encoding the FGFR1c kinase domain (residues 458-765; FGFR1K) was cloned into the pFastBac1 vector (Invitrogen, Carlsbad, CA, USA) using BamHI and HindIII restriction sites. The construct contained an N-terminal His$_6$ tag, and were verified in DNA sequencing analyses. Recombinant bacmids were then transfected into Sf9 cells using the Bac-to-Bac expression system (*Zhang et al., 2018*). Briefly, Sf9 cells were infected with recombinant baculovirus and expressed the recombinant protein. Transfected cells were harvested after 2 days by centrifugation at 3,000*g* and were then lysed in buffer A containing 20 mM Tris, pH 8.0, 400 mM NaCl, 10% glycerol, protease inhibitor cocktail (Roche, Basel, Switzerland), 10 ng/ml PMSF, 100 μM Na$_3$VO$_4$, and 1% triton X-100 on ice for 30 min. After centrifugation at 50,000 rpm for 45 min, proteins were purified using a Ni-NTA column and size exclusion chromatography (SEC) with a Superdex 200 GL 16/60 column (GE Healthcare, Chicago, IL, USA) at a flow rate of one ml/min at 4 °C. 20 mM Tris, pH 8.0, 400 mM NaCl was used as the running buffer. Target fractions were pooled and concentrated, finally stored at −80 °C.

## Preparation of NTA(Ni)-nanodiscs

To prepare nanodiscs, 95% DMPC and 5% DGS-NTA(Ni) powders were dissolved in buffer containing 10 mM potassium phosphate (pH 7.4) to prepare 20 mg/mL lipid stock solution. Subsequently, 22-amino acid peptides were dissolved in buffer containing 40 mM potassium phosphate, (pH 7.4), to make a 10 mg/mL peptide stock solutions. Peptide and lipids were then mixed at indicated volume ratios and incubated at 50 °C for 10 min and then at room temperature for 10 min. This procedure was repeated three to five times until the solution became clear, and three freeze and thaw cycles were then performed between −80 °C and room temperature to homogenize the nanodiscs (*Zhang et al., 2016*). The resulting nanodisc solutions were then characterized using SEC with a Superdex 200 GL 10/300 column (GE Healthcare, Chicago, IL, USA) at a flow rate of 0.5ml/min at room temperature. 20 mM PB, (pH 7.4), 50 mM NaCl was used as the running buffer. Target fractions were pooled and concentrated, prepared for kinase assays.

## Transmission electron microscopy

The specimens for transmission electron microscopy (TEM) was prepared by negative-stain technique. Carbon-coated copper grids (200 mesh) were cleaned for 5 min in plasma cleaner. Nanodisc samples were incubated on the grids for 1 min, and the excess was removed by blotting with filter paper. The grids were then stained with phosphorotungstic acid for 20 s and excess solution was removed by blotting. Air-dried samples were stored in a desiccator until observation. All TEM measurements were carried out using a FEI Tecnai Spirit tandem electron microscope (TEM; 100 kV). The particle diameter distribution and discoidal shape analysis were based on the 2D class averaging using EMAN2. For each peptide nanodiscs, at least 3,000 particles were selected and classified.

## Kinase assays

The activities of the kinase domains that were attached to nanodiscs were assayed by pre-incubating FGFR1K with nanodiscs of various diameters at 4 °C overnight. A Superdex 200 10/300 GL column was used to separate FGFR1K NTA(Ni)-nanodiscs from FGFR1K and empty nanodiscs prior to the start of kinase assays. Control samples of the kinase domain in solution without nanodiscs were also assayed using the same protocols. Kinase activity assays were then performed with 1.5 nM purified FGFR1K or FGFR1K NTA(Ni)-nanodiscs in buffer containing 5 mM ATP, 10 mM MgCl$_2$ in 20 mM Tris, and 200 mM NaCl (pH 8.0) at room temperature. Reactions were allowed to proceed at room temperature for indicated times and were then quenched using 50 mM EDTA. Subsequently, 6× loading buffer was added to each sample and proteins were loaded onto polyacrylamide-sodium dodecyl sulfate gels. After electrophoretic separation, proteins were transferred to nitrocellulose membranes, and these were then blocked using 5% BSA in TBS-T. Proteins were probed with various antibodies and the resulting blots were visualized using chemiluminescence autoradiography. The results were quantitated by ImageJ analysis. Data represent the average of at least four experiments. Statistical tests were conducted using either unpaired two-tailed Student's $t$-test or one-way analysis of variance with post hoc Dunnett's test. Error bars depict the S.D. $p$-Values of $^*p < 0.05$, $^{**}p < 0.01$, and $^{***}p < 0.001$ were considered to be statistically significant.

## Isothermal titration calorimetry

Isothermal titration calorimetry (ITC) measurements were performed on a MicroCalITC200 instrument (MicroCal GE, USA). FGFR1K proteins were diluted into 0.04 mM with the buffer of 20 mM PB, pH 7.4, 50 mM NaCl then titrated into the NTA(Ni)-Nanodiscs solution with the same buffer at 25 °C. All buffers and samples were centrifuged at 13,000 rpm for 30 min to minimize possible bubble formation during ITC stirrings. A control titration, consisting of the same titration buffer solution in the sample cell, was subtracted from each experimental titration to account for heat of dilution. The corrected binding isotherms were fitted to obtain the binding constant ($K$), the number of binding sites ($N$), enthalpy change ($\Delta H$) and entropy change ($\Delta S$), by using Malvern MicroCal PEAQ-ITC software.

# RESULTS

## Assembly of nanodiscs of various diameters

Membrane mimic systems are prerequisites for studies of the in vitro effects of membrane binding on FGFR1K. Herein, we assembled nanodiscs using the amphipathic 22-residue peptide PVLDLFRELLNELLEALKQKLK (22A) based on an amphipathic α-helical segment of the apolipoprotein A-I (Apo A-I) (*Zhang et al., 2016*). In contrast with the commonly used MSP from Apo A-I, which forms fix-sized nanodiscs, 22A allows assembly of size controllable nanodiscs using a comparatively simple procedure. We prepared nanodiscs of three different sizes by varying molar ratios of lipid (DMPC) to peptide. At the lipid/peptide ratios of 1:9, 1:3, and 1:1, single peaks were present on SEC elution

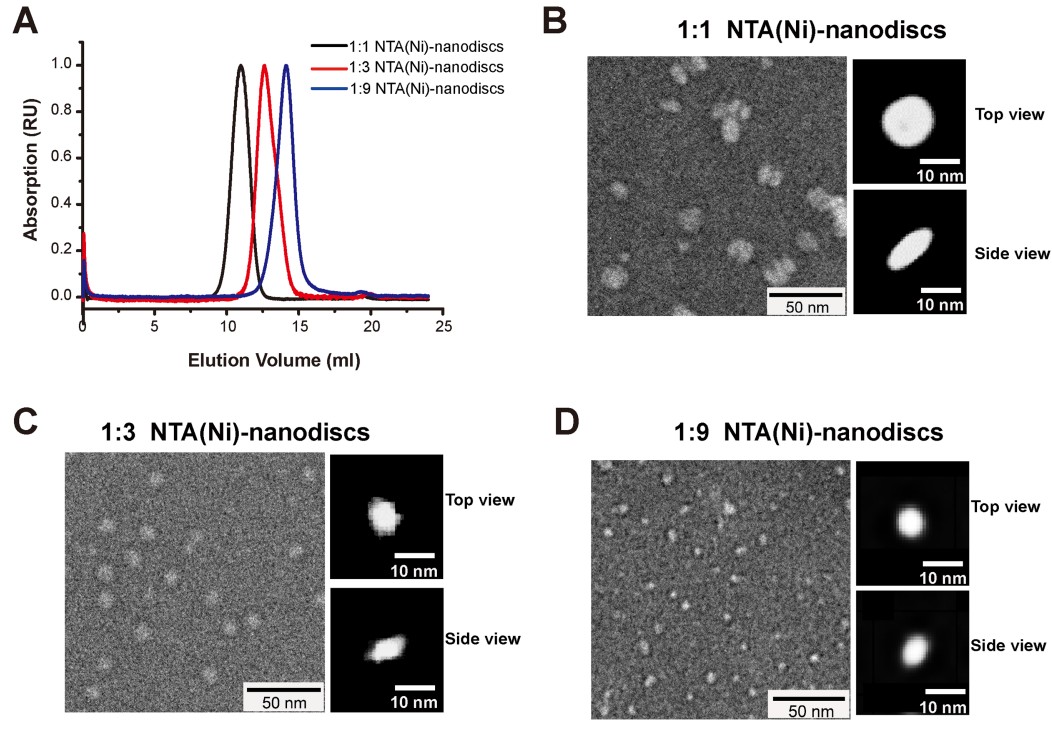

**Figure 2 Characterizations of peptide NTA(Ni)-nanodiscs.** (A) Size exclusion chromatography of peptide NTA(Ni)-nanodiscs with lipid:peptide ratios of 1:1 (black), 1:3 (red), and 1:9 (blue). (B) Negative stain EM images of 1:1 nanodisc particles (~12.0 nm). (C) Negative stain EM images of 1:3 nanodisc particles (~8.6 nm). (D) Negative stain EM images of 1:9 nanodisc particles (~6.3 nm). The top and side views of nanodiscs were selected from the two-dimensional classified nanoparticles (Fig. S2).

profiles at 14.1, 12.6, and 10.9 ml, each indicating single assembled forms with homogeneous size distributions (Fig. 2A). The elution peak position of peptide nanodisc of 1:3 was similar to that of the generally used MSPΔH5 nanodiscs, indicating similar nanodisc sizes (Fig. S1). TEM of 22A nanodiscs (Fig. 2B–2D) provide a direct evidence that peptide nanodiscs form disc-like assembly. The appearance of nanodisc stacking at the ratio of 1:1 (Fig. S2) was an artifact associated with negative staining sample treatment, which were also reported in other study (*Kumar et al., 2017*). The average diameters from 2D class average analysis are 6.3 ± 1.1, 8.6 ± 1.2, and 12.0 ± 1.8 nm, respectively. DLS analyses (Fig. S3) also verified that the diameters of assembled nanodiscs change linearly with lipid/peptide molar ratios, although the sizes derived are slightly smaller than those from TEM (Table S1). Such a discrepancy in sizes is likely caused by differences in sample treatment as well as data analysis.

## Chelating FGFR1K onto NTA(Ni)-nanodiscs

To attach the kinase to NTA(Ni)-nanodiscs and determine in vitro activity, we generated a FGFR1K (residues 458-765) construct with an N-terminal $His_6$ tag. The protein was expressed in a Sf9 cell expression system and was purified using Ni affinity chromatography followed by SEC (Fig. S4). Nanodiscs were then assembled using neutral DMPC lipid containing 5% NTA-Ni-DGS. $His_6$ tagged FGFR1K was incubated with various NTA

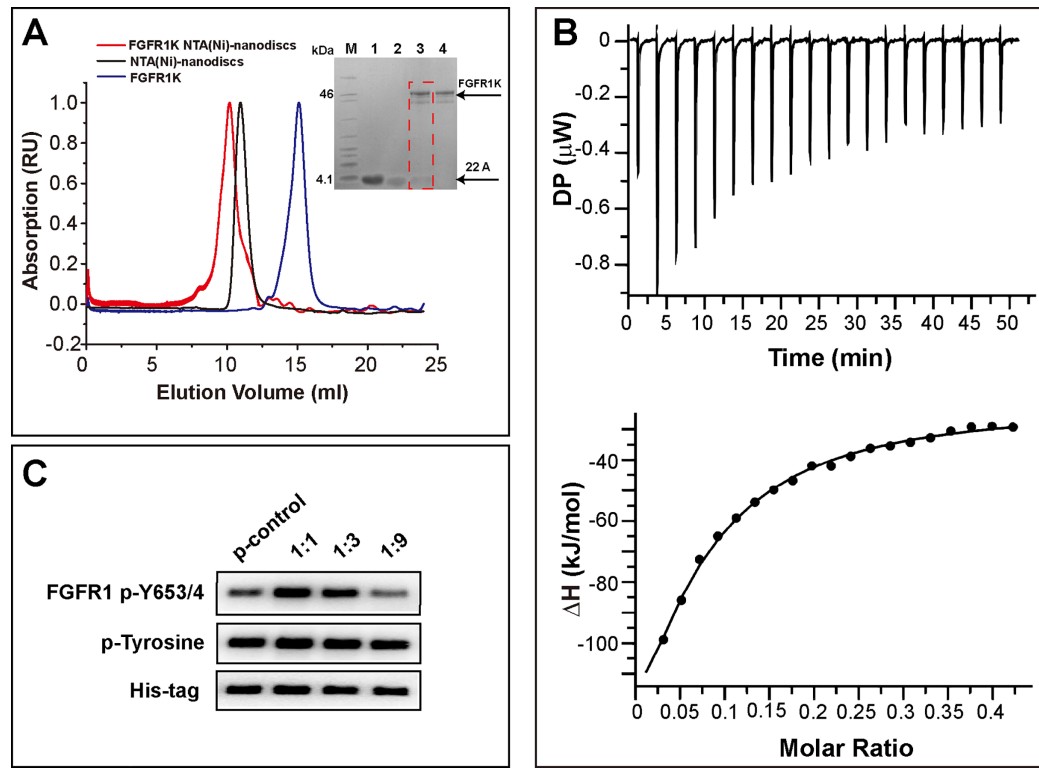

**Figure 3 Binding of FGFR1K to NTA(Ni)-nanodiscs.** (A) The size exclusion chromatography of FGFR1K with NTA(Ni)-nanodiscs (1:1) (red line with an elution peak at 10.2 ml), NTA(Ni)-nanodiscs (1:1) only (black line with an elution peak at 10.9 ml), and FGFR1K only (blue line with an elution peak at 15.1 ml). SDS PAGE was performed to verify the formation of FGFR1K- NTA(Ni) nanodisc complex (lane 3), with controls of peptide 22A (lane 1), NTA(Ni)-nanodiscs (lane 2), FGFR1K (lane 4) and protein makers (M). (B) The dissociation constant (KD) of FGFR1K to NTA(Ni)-nanodiscs is determined to be 2.4 μM according to the Isothermal titration calorimetry (ITC) measurements. To account for heat of dilution, a control titration of buffer solution into nanodisc solution (Fig. S5) was performed and subtracted from the titration of FGFR1K into nanodisc solution. (C) FGFR1K enzyme activities on peptide NTA(Ni)-nanodiscs of varying diameters. The total phosphorylation and phosphorylation levels of tyrosine 653/4 were detected using western blotting analyses with corresponding antibodies after 10-min reactions. The reaction of FGFR1K in solution was set as control.

(Ni)-nanodiscs overnight and the resulting complexes were collected using SEC. The purified FGFR1K possessed enzyme activity, and the phosphorylation happened at tyrosine sites can be monitored using specific antibodies (Fig. S6). The His-tagged FGFR1K can bind to NTA(Ni)-nanodiscs (1:1) as indicated in the SEC profile of Fig. 3A, where a higher molecular weight complex forms and therefore elutes faster (red line with an elution peak at 10.2 ml) than NTA(Ni)-nanodiscs (1:1) only (black line with an elution peak at 10.9 ml) sample and FGFR1K only sample (blue line with an elution peak at 15.1 ml). The co-presence of FGFR1K and 22A peptide bands in SDS PAGE confirms the binding of FGFR1K to NTA(Ni)-nanodiscs. FGFR1K and peptide NTA(Ni)-nanodisc forms stable complex with a dissociation constant of 2.4 μM as determined by the ITC experiments (Fig. 3B). The effects of membrane binding were characterized by monitoring total tyrosine phosphorylation and that of Y653/4 in the A-loop of FGFR1K (Fig. 3C).
The total tyrosine phosphorylation and the phosphorylations of Y653/4 in the presence of NTA(Ni)-nanodiscs of 1:3 and 1:9 are less or comparable to the aqueous control. Among the three peptide nanodiscs assembled, the highest autophosphorylation activity of FGFR1K was achieved with nanodisc of 1:1. According to the cross-linking results, the highest FGFR1K oligomer to monomer ratio (Fig. S7) also appears in the presence of nanodisc (1:1), suggesting a correlation between FGFR1K autophosphorylation activity and nannodisc sizes.

### Kinetic analysis of FGFR1K autophosphorylation on NTA(Ni)-nanodiscs

To characterize the autophosphorylation kinetics, we monitored phosphorylation over time using anti-p-tyrosine antibodies (Fig. S8). As shown in Fig. 4A, the total phosphorylation plateaued at 1 h, and membrane binding significantly enhanced (twofold) tyrosine autophosphorylation. The phosphorylation of the A-loop residues Y653 and Y654, which has been reported to be critical for upregulation of kinase activity (*Turner & Grose, 2010*), were also monitored by western blotting analyses using antibodies against p-Y653 and p-Y654. In aqueous solution, minor changes were observed for Y653 phosphorylation levels over time, whereas the presence of nanodiscs led to a maximal 10-fold increase in phosphorylation of Y653 over 2 h, which is fivefold greater than that in aqueous controls (Fig. 4B). It takes only a few minutes for Y654 to reach a phosphorylation plateau in aqueous solution. In the presence of peptide nanodiscs, the phosphorylation of Y654 continues to grow, and the phosphorylation at 2 h is about fourfold compared with the aqueous control (Fig. 4C). The Y654 phosphorylation increases 17 times comparing to its initial phosphorylation level. The enhancements of both Y653 and Y654 phosphorylation strongly indicate that, the binding of FGFR1K on 2D membrane nanodiscs facilitates Y653 and Y654 activation.

## DISCUSSION

Tyrosine autophosphorylation of RTKs plays a critical role in the regulation of kinase activities and in the recruitment and activation of downstream substrates. Many studies of kinase-only domains in aqueous environments, nevertheless, missed the conditions provided by native membrane surface that is essential for autophosphorylation. For example, autophosphorylation requires the kinase domains to orient and cooperate, respectively, on dimer interfaces, differing considerably from the random collisions that occur in aqueous environments. However, the hydrophobicity associated with the trans-membrane fragment of full-length FGFR hampers protein expression and purification, and subsequent biophysical and biochemical analyses. These conditions necessitate mimicry of cell membrane environments to promote receptor dimerization and cooperative cross-receptor interactions.

Nanodiscs provide nanoscale surfaces on which functional complexes can be assembled with lipid membranes, offering a good membrane mimic system for investigating membrane-mediated molecular recognition events. In contrast with fixed-sized nanodiscs that are assembled with MSP, the present 22A-peptide based nanodisc assembly allows easy and accurate adjustments of nanodisc sizes by changing lipid/peptide ratios. Further addition of nickel chelating lipid allowed the anchoring of N-terminal His$_6$ tagged

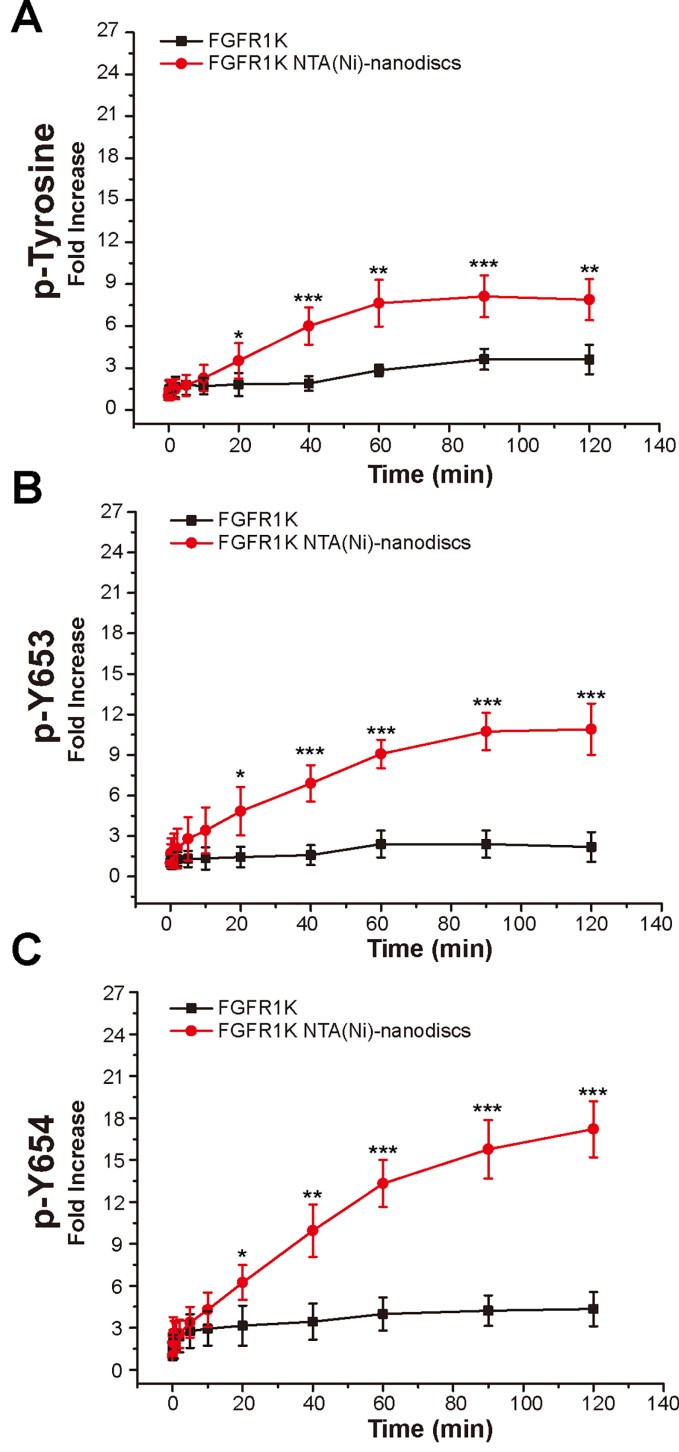

**Figure 4 Autophosphorylation kinetics of FGFR1K in solution and on NTA(Ni)-nanodiscs.** The relative phosphorylation levels of the total tyrosine phosphorylation (A), p-Y653 (B), and p-Y654 (C) of FGFR1K in solution (black lines) and on NTA(Ni)-nanodiscs (red lines) at indicated time points were calculated based on western blotting data (Fig. S6). The increment fold of phosphorylation at specific time point is defined by the intensity of the western blotting band relative to that of the starting point. Specific antibodies of phospho-tyrosine, FGFR1 p-Y653, and FGFR1 p-Y654 were used for western blotting analysis. $n > 3$; $*p < 0.05$, $**p < 0.01$, and $***p < 0.001$.

FGFR1K onto lipid membrane even in the absence of FGFR transmembrane domain. These protein-nanodisc assemblies led to relatively well-defined samples that are amenable to various analyses, including DLS, TEM, and SEC. This approach also has promise in other systems involving membrane associated proteins.

Fibroblast growth factor receptors transduce biochemical signals via lateral dimerization and autophosphorylation on plasma membranes. To this end, the kinase domain of one FGFR molecule serves as an active enzyme, and the other, its substrate. In aqueous solution, although kinase domains diffuse freely and conveniently cross-linked (Fig. S7), the phosphorylation level is rather low (Fig. 3C), and occurs only when two kinase domains diffuse and collide with appropriate orientation relative to each other. The presence of NTA(Ni)-nanodiscs, on the other hand, restricts kinase domains to diffuse in a 2D space, and increases the chances of kinase collision and reaction. Moreover, the membrane anchoring also helps in coordinating the kinase enzyme and kinase substrate, into the appropriate orientations for phosphorylation. From our data, only NTA(Ni)-nanodiscs at a lipid:peptide ratio of 1:1 exhibit significant enhancements of kinase oligomerization and activation, indicating the necessary of sufficient membrane surface for autophosphorylation. The flexibility and convenience of peptide nanodiscs in tuning membrane areas, therefore make it a versatile membrane mimic system in studying various membrane associated phenomena.

Autophosphorylation of the tyrosine residues of FGFR1K is a precisely ordered process and is kinetically driven, as shown in mass spectroscopy analyses (*Lew et al., 2009*). These studies suggest that each step of the process is accompanied by a different orientation of catalytic kinase relative to substrate kinase. FGFR phosphorylation kinetics on the membrane is likely more complicated, warranting further studies to determine the order of phosphorylation reactions on membranes, and how membranes affect phosphorylation kinetics. Although the present western blotting analyses of the kinetics of autophosphorylation at tyrosine sites on nanodiscs are semi quantitative, our nanodiscs significantly increased phosphorylation levels of all tyrosine residues and those at Y653 and Y654. Moreover, in the presence of nanodiscs, Y653 phosphorylation plateaued after 1 h, whereas the Y654 phosphorylation continue to increase (Fig. 4), indicating phosphorylation of Y653 happens earlier than that of Y654. This is consistent with previous report that Y653 is the first site to be phosphorylated and stimulates kinase activity by 50- to 100-fold (*Lew et al., 2009*). However, due to the complexity of the autophosphorylation mechanism and the lack of available antibodies for specific phosphor-tyrosine forms, we were not able to determine the order of autophosphorylation precisely. The regulatory effects of membranes on autophosphorylation at various tyrosine sites may be better characterized using techniques such as mass spectroscopy analysis.

## CONCLUSIONS

In conclusion, we developed an NTA(Ni)-nanodiscs system for which the nanodisc size can be finely adjusted. An N-terminal His$_6$-tagged FGFR1K anchored spontaneously onto the membrane with orientations that mimic those in plasma membranes. The 2D membrane surfaces provided by peptide nanodiscs significantly enhanced the

phosphorylation rates of Y653 and Y654 in FGFR1K A-loop. The present system has potential applications to in vitro studies of other membrane associated phenomena.

### Funding
This work was supported by the grants from the National Natural Science Foundation of China (Grant No. U1532269, 31800645) and the Major Program of Development Foundation of Hefei Center for Physical Science and Technology (Grant no. 2017FXZY004). The funders had no role in study design, data collection and analysis, decision to publish, or preparation of the manuscript.

### Grant Disclosures
The following grant information was disclosed by the authors:
National Natural Science Foundation of China: U1532269, 31800645.
Major Program of Development Foundation of Hefei Center for Physical Science and Technology: 2017FXZY004.

### Competing Interests
The authors declare that they have no competing interests.

### Author Contributions
- Juanjuan Liu performed the experiments, analyzed the data, prepared figures and/or tables, authored or reviewed drafts of the paper, approved the final draft.
- Lei Zhu conceived and designed the experiments, analyzed the data, authored or reviewed drafts of the paper, approved the final draft.
- Xueli Zhang performed the experiments, contributed reagents/materials/analysis tools.
- Bo Wu analyzed the data, approved the final draft.
- Ping Zhu contributed reagents/materials/analysis tools.
- Hongxin Zhao conceived and designed the experiments, performed the experiments, analyzed the data, prepared figures and/or tables, authored or reviewed drafts of the paper, approved the final draft.
- Junfeng Wang conceived and designed the experiments, authored or reviewed drafts of the paper, approved the final draft.

### Data Availability
The raw measurements are available in the Supplemental Files.

### Supplemental Information
Supplemental information for this article can be found online at http://dx.doi.org/10.7717/peerj.7234#supplemental-information.

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
