# Peer review of "Peptide-based NTA(Ni)-nanodiscs for studying membrane enhanced FGFR1 kinase activities"

_PeerJ, doi:10.7717/peerj.7234_

## Round 0.1 · original submission · Minor Revisions

Dear Junfeng,

Thank you for giving us the opportunity to reconsider your revised manuscript and thank you for your patience while your manuscript has been reviewed and please accept my apologies for the delay in responding.

Your paper has been seen by two expert reviewers (one of which was a reviewer of the previous submission) and overall, they are supportive of publication. However reviewer#2 has raised a number of minor concerns (which are detailed below) that will need to be satisfactorily addressed.

Thank you for the opportunity to consider this work, and please do not hesitate to contact me in case you should have any additional question regarding this decision or the reports. I look forward to your revision.

Yours sincerely,

Alexis Verger
Academic Editor, PeerJ

Reviewer 1 ·

Basic reporting

No comment.

Experimental design

No comment.

Validity of the findings

No comment.

Additional comments

The authors have sufficiently responded to my questions.

Reviewer 2 ·

Basic reporting

More reference on the background of nanodiscs could be cited, e.g. Assembly of phospholipid nanodiscs of controlled size for structural studies of membrane proteins by NMR, which was published in Nature Methods.

Experimental design

no comment

Validity of the findings

Average diameters of the nanodiscs determined from TEM data should be based on 2D class average results. However, no 2D class average results are shown in the article.

Additional comments

1. The authors used DMPC as the main lipid for nanodisc formation. However, DMPC is an artificial lipid and is shorter by two carbons than the shortest native lipid. Could the authors comment on the reason of using DMPC and how physiologically relevant this study is by using DMPC?
2. Line 170: the average diameters of the nanodiscs by TEM should be based on 2D class averages, the authors should provide the 2D class average results in the figure; in additon, it seems in Figure 2B that the nanodiscs are stacking onto each other for 1:1 ratio, while they are well separated from each other for 1:3 and 1:9 ratios. Does the authors have an explaination on this observation? How does this affect the phosphorylation? In Line 190-196, the authors stated that phosphorylation is correlated with the size of the nanodiscs based on the observation that the 1:1 ratio discs provided highest phosphorylation. Is it possible that phosphorylation is related to the stacking effect instead of the size of the nanodiscs?
3. Line 184-186: the black should be nanodiscs only and the red should be the complex, please make sure the text matches the data in the figure; in addition, the SEC profile of FGFR1K alone should also be shown in Figure 3A for better comparison.
4. Line 186-188: the authors performed ITC experiments to measure binding affinity between empty nanodiscs and FGFR1K. ITC experiments on nanodiscs are hard to perform due to bubble formation during stirring in the ITC cell. The authors should indicate in the experimental section on how to avoid air bubble formation during ITC experiments. In additon, proper control experiments are very important for ITC results to be reliably interpreted. The authors should include experimental design of the control experiments in the experimental section and the results of the control in the supplementary material.
5. More evidence of complex formation between nanodiscs and FGFR1K is needed in addition to ITC results. SDS PAGE gel for the peak in Figure 3A could be presented to show the presence of both FGFR1K and 22A (if visible on gel). Or TEM images could be presented for this peak to show that FGFR1K is associated with nanodiscs.

---

## Round 0.2 · accepted · Accept

I confirm that the authors carefully took into account the reviewer's comments and significantly improved the manuscript. The current version of the article is suitable for publication in PeerJ.